# Reproducibility Study of "Counterfactual Generative Networks"

## Reproducibility Summary

**Scope of Reproducibility**

In this work, we study the reproducibility of the paper *Counterfactual Generative Networks* (CGN) by Sauer and Geiger to verify their main claims, which state that (i) their proposed model can reliably generate high-quality counterfactual images by disentangling the shape, texture and background of the image into independent mechanisms, (ii) each independent mechanism has to be considered, and jointly optimizing all of them end-to-end is needed for high-quality images, and (iii) despite being synthetic, these counterfactual images can improve out-of-distribution performance of classifiers by making them invariant to spurious signals.

**Methodology**

The authors of the paper provide the implementation of CGN training in PyTorch. However, they did not provide code for all experiments. Consequently, we re-implemented the code for most experiments, and run each experiment on 1080 Ti GPUs. Our reproducibility study comes at a total computational cost of 112 GPU hours.

**Results**

We find that the main claims of the paper of (i) generating high-quality counterfactuals, (ii) utilizing appropriate inductive biases, and (iii) using them to instil invariance in classifiers, do largely hold. However, we found certain experiments that were not directly reproducible due to either inconsistency between the paper and code, or incomplete specification of the necessary hyperparameters. Further, we were unable to reproduce a subset of experiments on a large-scale dataset due to resource constraints, for which we compensate by performing those on a smaller version of the same dataset with our results supporting the general performance trend.

**What was easy**

The original paper provides an extensive appendix with implementation details and hyperparameters. Beyond that, the original code implementation was publicly accessible and well structured. As such, getting started with the experiments proved to be quite straightforward. The implementation included configuration files, download scripts for the pretrained weights and datasets, and clear instructions on how to get started with the framework.

**What was difficult**

Some of the experiments required severe modifications to the provided code. Additionally, some details required for the implementation are not specified in the paper or inconsistent with the specifications in the code. Lastly, in evaluating out-of-distribution robustness, getting the baseline model to work and obtaining numbers similar to those reported in the respective papers was challenging, partly due to baseline model inconsistencies within the literature.

**Communication with original authors**

We have reached out to the original authors to get clarifications regarding the setup of some of the experiments, but unfortunately, we received a late response and only a subset of our questions was answered.

Submitted to ML Reproducibility Challenge 2020. Do not distribute.

# 1 INTRODUCTION

Despite the considerable popularity of deep learning models within the field of artificial intelligence, recent literature suggests that these networks have a tendency of learning simple correlations that perform well on a benchmark dataset, instead of more complex relations that generalize better [1, 17, 21]. This phenomenon, which is referred to as shortcut learning by Geirhos et al. [10], makes these models more sensitive to input perturbation and unseen input contexts.

In order to enhance the robustness and interpretability of classifiers, Sauer and Geiger [22] introduce the idea of a *Counterfactual Generative Network* (CGN). Using appropriate inductive biases to disentangle separate components within the input images, such as object shape, object texture, and background, this model is capable of augmenting training data with generated counterfactual images. The authors claim that, using this model, they were able to improve out-of-distribution (OOD) robustness with only a marginal performance decrease for the original classification task.

In this work, we aim to reproduce their findings, verify their claims, and perform additional experimental results to provide further evidence to support their claims. In summary, this work makes the following contributions:

- We reproduce the main experiments conducted by Sauer and Geiger [22] to identify which parts of the experimental results supporting their claims can be reproduced, and at what cost in terms of resources (e.g., computational cost, development effort, and communication with the authors).
- We improve the performance consistency of the CGN during training.
- We extend upon the work of Sauer and Geiger by empirically analyzing the decisions made by classifiers based on their proposed model. Based on this analysis, we propose a method to quantify the robustness of such classifiers against spurious correlations.

## 1.1 Scope of Reproducibility

Distinguishing between spurious and causal correlation is an active topic in causality research [15, 18]. One central principle in causal inference is the assumption of independent mechanisms (IMs), which states that a causal generative process is composed of autonomous modules that do not influence each other [19, 22, 24]. The CGN introduced in the original paper exploits this idea to decompose the image generation process into three IMs, each controlling one factor of variation (FoV), namely the shape, texture, and background. Using this, the authors take a step towards more robust and interpretable classifiers that explicitly expose the causal structure of the classification task. In this reproducibility study, our main goal is to verify the following claims of the original paper:

- **High-Quality Counterfactuals (HQC)**: By exploiting proper inductive biases, the CGN is able to reliably learn the independent mechanisms, which allow for the generation of high-quality counterfactual images by disentangling the shape, texture and background of the image.
- **Inductive Bias Requirements (IBR)**: Each independent mechanism has to be considered, and jointly optimizing all of them end-to-end is needed for high-quality images.
- **Out-of-Distribution Robustness (ODR)**: Despite being synthetic, the counterfactual images can improve out-of-distribution performance of classifiers by making them invariant to spurious signals.

The remainder of this work is structured as follows. In Section 2, we introduce the model proposed in the original paper to provide the reader with the required background knowledge. Section 3 then summarizes our approach to reproduce the original paper. Subsequently, Section 4 presents the replicated results and compares them to the original paper. Section 5 concludes this work by discussing our experience with reproducing the research by Sauer and Geiger [22].

# 2 COUNTERFACTUAL GENERATIVE NETWORK

The counterfactual generative network is a manifestation of a structural causal model (SCM) for the task of image classification [22]. It decomposes the image generation process into four IMs whose losses are jointly optimized in an end-to-end matter. An overview of the CGN architecture is shown in Appendix A.

**Shape mechanism:** The shape mechanism $f_{shape}$ captures the shape as a binary mask $m$, where 1 corresponds to the object and 0 to the background. For this purpose, it first samples a pre-mask $\tilde{m}$ with exaggerated features from a

fine-tuned BigGAN [4], and extracts the binary mask using a pretrained U2-Net [20]. The shape loss $\mathcal{L}_{shape}$ comprises (1) the *pixelwise binary entropy* of the mask, and (2) the mask loss:

$$\mathcal{L}_{mask}(\boldsymbol{m}) = \mathbb{E}_{p(\boldsymbol{u},y)} \left[ \max\left(0, \tau - \frac{1}{N}\sum_{i=1}^{N} m_i\right) + \max\left(0, \frac{1}{N}\sum_{i=1}^{N} m_i - \tau\right) \right]. \tag{1}$$

The pixelwise binary entropy forces the output to be close to either 0 or 1, whereas the mask loss discourages trivial solutions that are outside the interval defined by $\tau$.

**Texture mechanism:** The texture mechanism $f_{text}$ generates the texture of the object. For MNIST, Sauer and Geiger use an additional layer that divides its input into patches and randomly rearranges them. In contrast, for ImageNet, they sample patches from the regions where the mask values are the highest and concatenate them into a patch grid $\boldsymbol{pg}$. This mechanism is optimized by minimizing the perceptual loss between the foreground $\boldsymbol{f}$ and the patch grid $\boldsymbol{pg}$. As such, the background gradually transforms into object texture during training.

**Background mechanism:** The background mechanism $f_{bg}$ models the background $\boldsymbol{b}$ of the image. It removes the object from the output of the BigGAN backbone and inpaints it using U2-Net by *minimizing* the predicted saliency. Because there is no need for a globally coherent background in the MNIST setting, the MNIST variant of the CGN includes a second texture mechanism rather than a dedicated background mechanism.

**Composer:** The composer $C$ combines the output of the aforementioned mechanisms into a single composite image

$$\boldsymbol{x}_{gen} = C(\boldsymbol{m}, \boldsymbol{f}, \boldsymbol{b}) = \boldsymbol{m} \odot \boldsymbol{f} + (1 - \boldsymbol{m}) \odot \boldsymbol{b}, \tag{2}$$

where $\boldsymbol{m}$ is the mask, $\boldsymbol{f}$ is the foreground, $\boldsymbol{b}$ is the background, and $\odot$ is the Hadamard product. To optimize this mechanism, Sauer and Geiger use an external conditional GAN (cGAN) that generates pseudo-ground-truth images $\boldsymbol{x}_{gt}$ from the same noise $\boldsymbol{u}$ and label $y$ that is fed into the aforementioned mechanisms of the CGN. Using this, they minimize the reconstruction loss $\mathcal{L}_{rec}$ between the composite image $\boldsymbol{x}_{gen}$ and the pseudo-ground-truth image $\boldsymbol{x}_{gt}$.

During training, each independent mechanism learns a class-conditional distribution over shapes, textures, or backgrounds. It can then generate counterfactual images by randomizing the noise $\boldsymbol{u}$ and label input $y$ for each mechanism. A more detailed explanation regarding the purpose of these counterfactual images and the connection with explainable artificial intelligence (XAI) can be found in Appendix B.

In order to encode invariance to spurious correlations, Sauer and Geiger train classifiers on generated counterfactual data that retain the label from the shape with randomized texture and backgrounds. For MNISTs, they use a standard CNN feature extractor followed by a single classification head. For ImageNet on the other hand, they use a CNN backbone with three classifier heads: shape, texture, and background; each invariant to all but one factor of variation. The final prediction is obtained by averaging the individual head predictions.

# 3 METHODOLOGY

The original implementation of the CGN is publicly available [23], but most of the experiments conducted in the original paper to support their claims are not. Consequently, we use the authors' code for the implementation of the CGN, and re-implement the experiments and relevant evaluation metrics based on the descriptions provided in the paper. Furthermore, we both improve and extend upon the work of Sauer and Geiger by providing additional experiments and results. Because a description of the GAN used in the original paper was not provided, we use a DCGAN [14].

## 3.1 Datasets

The experiments conducted in the original paper involve two tasks, namely generating counterfactual examples and training a classifier to be invariant to spurious correlations. We follow the paper and reproduce their evaluations on multiple datasets for each task. For both tasks, we present the relevant datasets and their main purpose in Table 1. Due to resource constraints, running all experiments on full ImageNet (IN-1k) is infeasible. As a compromise, we use ImageNet-mini (IN-Mini) [7], a small-scale variant of ImageNet. Although this dataset contains fewer samples, we found it to be sufficient to reproduce the main findings of the original paper and verify their claims. Moreover, this dataset includes the same classes as IN-1k and hence does not induce any decrease in difficulty of the classification task.

Table 1: **Datasets overview.** The datasets used for empirical evaluations across two tasks.

| Task | Datasets | Number of samples | | | Classes | Description | URL |
|---|---|---|---|---|---|---|---|
| | | Train | Test | Total | | | |
| Generating counterfactual samples | C-MNIST [2] | 50k | 10k | 60k | 10 | Foreground colour as a spurious correlation | Link |
| | DC-MNIST | 50k | 10k | 60k | 10 | Fore/background colour as spurious correlations | NA[1] |
| | W-MNIST | 50k | 10k | 60k | 10 | In-the-wild background with texture colour | NA[1] |
| | IN-1k [6] | 1M | 100k | 1.2M | 1000 | Large-scale evaluation | Link |
| | IN-mini [7] | 35k | 4k | 39k | 1000 | Small-scale evaluation | Link |
| Training invariant classifiers | MNISTs | 50k | 10k | 60k | 10 | Test different granularities of invariance | Link |
| | Cue-conflict [8] | NA | 1280 | 1280 | 16 | Tests shape-texture disentanglement | Link |
| | IN-9 variants [29] | ∼45k | ∼4k | ∼50k | 9 | Tests background-invariance | Link |

## 3.2 Hyperparameters

In order to match the original experiments as closely as possible, we used the same hyperparameters as the authors of the original paper whenever they were specified in the article. If the required hyperparameters for the experiments were not mentioned in the original paper, we relied on the default parameters given in the configuration files of the original implementation. In this case, we assume that these default parameters correspond to the parameters used for the described experiments.

## 3.3 Experimental setup and evaluation metrics

Our experimental setup is largely based on the description provided by Sauer and Geiger [22]. To that end, we will address claim HQC by performing a qualitative analysis on both MNIST and ImageNet. To verify claim IBR, we perform a loss ablation study in which we disable one loss at a time. Lastly, to address the main claim of the paper, namely ODR, we conduct a number of experiments on both MNIST and ImageNet to evaluate both out-of-distribution performance and spurious signal invariance of the invariant classifiers.

To provide further evidence to support claim ODR, we conduct additional experiments to visually explain the decisions made by the invariant classifiers based on gradient-based localization. For this purpose, we use a PyTorch implementation of GradCAM [11, 25], a class activation map method that weighs the 2D activations by the average gradient [25]. This method allows us to visualize the salient features on which the invariant classifiers base their predictions.

## 3.4 Computational requirements

We perform all experiments on a cluster whose nodes are equipped with Nvidia GeForce GTX 1080 Ti GPUs. Due to constraints in resources, we run most experiments once. As such, our experiments are indicative and not conclusive. Our reproducibility study comes at a total computational cost of 112 GPU hours (see Appendix D for more details).

## 4 EXPERIMENTAL RESULTS

### 4.1 Reproducibility study

**Evaluating counterfactual samples**    To verify claim HQC, we qualitatively evaluate counterfactual (CF) samples generated using CGN models on each dataset. For all our reproducibility experiments, we use the available pretrained weights for CGN to generate CFs. We found inconsistencies while training the CGN from scratch and refer the reader to Section 4.2.1 for a deeper investigation. For both MNIST and ImageNet, our results indicate that the quality of the generated CFs matches with the quality of those reported in the original paper, as shown in Figure 1 and Figure 2 respectively. For ImageNet, although we can easily recognize the FoVs in the generated CFs, they are highly unrealistic.

---

[1] This variant of MNIST is generated by the authors themselves and can be generated using their repository.

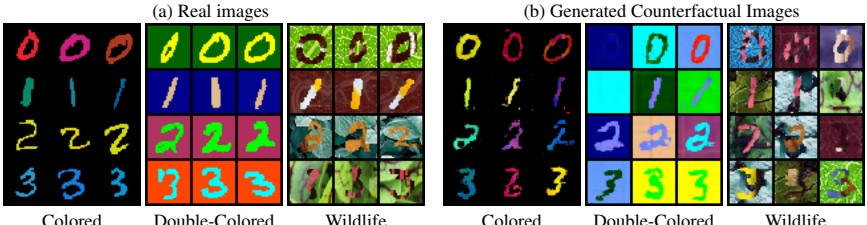

(a) Real images                                    (b) Generated Counterfactual Images

Colored    Double-Colored    Wildlife          Colored    Double-Colored    Wildlife

Figure 1: **Qualitative Analysis MNIST.** Left: Samples drawn from the different MNIST variations. Right: Counterfactuals generated by the CGN on MNIST variants. Notice that the CGN generates varying shapes, colors, and textures.

| | | | | | |
|---|---|---|---|---|---|
| Shape | Racer | Trench coat | Turtle | Vase | Malinois | Barrel |
| Texture | Clock | Cab | Cauliflower | Elephant | Viper | Piggy bank |
| Background | Toucan | Coral reef | Mushroom | Alp | Spider | Ibex |

Figure 2: **Qualitative analysis ImageNet.** Counterfactuals generated by the CGN on ImageNet.

**Evaluating loss ablation**   We attempt to reproduce the loss ablation study to verify claim IBR. The authors claim that a CGN can be trained from scratch within 12 hours on a GTX 1080Ti GPU. However, when running the experiments as described by the authors, the estimated training time exceeded 200 hours. Upon further inspection, we found an alternative configuration file containing the hyperparameters the authors used to train the CGN that was inconsistent with the default hyperparameters. Using these alternative hyperparameters, we managed to decrease the training time to approximately 20 hours. While the inception score magnitude directly depends on the number of generated images used for the calculation, the original paper did not specify the exact number of images used during the experiment. We empirically found that using 2000 images provides inception scores that resemble those reported in the original paper.

The results in Table 2 indicate that the inception scores follow a similar trend as reported by the authors (marked as x ). However, when disabling the texture loss, we found $\mu_{mask}$ to be 0.4, whereas the original paper reported a value of 0.9. This is a crucial difference, because the value of 0.9 of the original paper indicates a mask collapse, which the authors use to support claim IBR. Nonetheless, we were able to support this claim by performing an additional qualitative experiment. Specifically, if we look at some samples as shown in Appendix E, it is clear that the generated texture still includes some background. This indicates that the independent mechanisms for texture and background are no longer disentangled, which shows that the texture loss is indeed necessary.

**Evaluating invariant classifiers**   We perform a number of experiments to verify claim ODR. Specifically, we quantitatively evaluate the extent to which invariance is encoded in classifiers trained on CF data against those trained on original data. We also evaluate classifiers trained on vanilla GAN-generated data as a baseline. Since the vanilla GAN implementation was not provided in the released code, we implement it ourselves and refer the reader to Appendix F for details and generated samples.

Table 2: **Loss Ablation Study.** We turn off one loss at the time.

| $\mathcal{L}_{shape}$ | $\mathcal{L}_{text}$ | $\mathcal{L}_{bg}$ | $\mathcal{L}_{rec}$ | IS ⇑ | | $\mu_{mask}$ | |
|---|---|---|---|---|---|---|---|
| ✗ | ✓ | ✓ | ✓ | 100.8 | 85.9 | 0.3 | 0.2 |
| ✓ | ✗ | ✓ | ✓ | 186.5 | 198.4 | 0.4 | 0.9 |
| ✓ | ✓ | ✗ | ✓ | 200.9 | 195.6 | 0.1 | 0.1 |
| ✓ | ✓ | ✓ | ✗ | 19.3 | 38.4 | 0.4 | 0.3 |
| ✓ | ✓ | ✓ | ✓ | 156.1 | 130.2 | 0.3 | 0.3 |
| BigGAN (Upper Bound) | | | | 202.9 | | - | |

Table 3: **MNIST classification.** In the test-set, the texture and background are randomized; only the digits shape corresponds to the class.

| Setting | C-MNIST | | DC-MNIST | | W-MNIST | |
|---|---|---|---|---|---|---|
| | Train ⇑ | Test ⇑ | Train ⇑ | Test ⇑ | Train ⇑ | Test ⇑ |
| O(riginal) | 99.7 \| 99.5 | 37.6 \| 35.9 | 100 \| 100 | 10.5 \| 10.3 | 100 \| 100 | 10.8 \| 10.1 |
| GAN | 99.6 \| 99.8 | 32.5 \| 40.7 | 100 \| 100 | 10.6 \| 10.8 | 99.9 \| 100 | 11.2 \| 10.4 |
| CGN | 99.4 \| 99.7 | 92.3 \| 95.1 | 94.8 \| 97.4 | 86.5 \| 89.0 | 95.5 \| 99.2 | 81.4 \| 85.7 |
| O + GAN | 99.6 \| 99.8 | 41.5 \| 40.7 | 100 \| 100 | 10.0 \| 10.8 | 100 \| 100 | 11.1 \| 10.4 |
| O + CGN | 99.2 \| 99.7 | 95.9 \| 95.1 | 96.9 \| 97.4 | 85.5 \| 89.0 | 96.8 \| 99.2 | 62.8 \| 85.7 |

On MNIST variants, we identify an inconsistency in the experimental setup stated in the paper and code. The paper seems to suggest using a combination of original and CF dataset, but the code only uses CF data. As reported in Table 3, we experiment with both and observe similar results for C-MNIST and DC-MNIST. Surprisingly, for CGN, adding original data hurts the performance for W-MNIST (62.9 vs. 81.4). Apart from that, the majority of our results are within 5% variation from those reported in the paper (marked as x ), which supports the broader claim of better generalization even in the presence of spurious correlations (e.g., texture in case of colored MNIST).

To evaluate the invariance in classifier heads on IN-mini, we first reproduce the experiment regarding shape bias from the original paper. The shape bias is defined as the fraction of test samples for which the predicted label matches the shape label of the input image [8]. In this case, we evaluate labels with predictions from each head. As reported in Table 4, our results are smaller in comparison to the IN-1k results reported in the original paper. Nonetheless, the overall trend does support claim ODR. Additionally, we replicate the experiment regarding the evaluation of background robustness. The paper uses the notion of BG-gap that measures classifiers' reliance on background signal [28]. Our results, shown in Table 5, again slightly deviate from the original paper but the trend supports claim ODR.

Table 4: **Shape vs. texture.** Evaluation of shape biases of independent classifiers.

| Trained on | Shape Bias | top-1 ⇑ | top-5 ⇑ |
|---|---|---|---|
| IN + GCN/Shape | 54.8 | | |
| IN + GCN/Text | 16.7 | 74.0 | 91.7 |
| IN + GCN/Bg | 22.9 | | |
| IN-mini + GCN/Shape | 49.1 | | |
| IN-mini + GCN/Text | 20.5 | 56.2 | 79.1 |
| IN-mini + GCN/Bg | 25.7 | | |

Table 5: **Backgrounds Challenge.** Evaluation of robustness against adversarially chosen backgrounds.

| Trained on | IN-9 ⇑ | Mixed-Same ⇑ | Mixed-Rand ⇑ | BG-Gap ⇓ |
|---|---|---|---|---|
| IN | 95.6 | 86.2 | 78.9 | 7.3 |
| SIN | 89.2 | 73.1 | 63.7 | 9.4 |
| IN + SIN | 94.7 | 85.9 | 78.5 | 7.4 |
| Mixed-Rand | 73.3 | 71.5 | 71.3 | 0.2 |
| IN + CGN | 94.2 | 83.4 | 80.1 | 3.3 |
| IN-mini + CGN | 86.8 | 73.2 | 68.3 | 4.9 |

To evaluate the effect of using more counterfactual datapoints or generating more counterfactual images per sampled noise, Sauer and Geiger performed an MNIST Ablation Study in the original paper. Our reproduction for this experiment, along with a more detailed description regarding the experiment and results, can be found in Appendix G.

## 4.2 Results beyond original paper

### 4.2.1 Improving CGN training on MNISTs

While training the CGN on the MNIST, we encountered an issue that was not mentioned in the original paper. During the training process, we observed that the digit masks had a tendency of collapsing to an erroneous state, from where the digits would no longer improve during training. For this reason, it was not possible for us to reproduce the CGN training on the MNIST data using the default configuration. Therefore, we have proposed a solution that makes the CGN training on the MNIST datasets more consistent. Details regarding our solution can be found in Appendix C.

### 4.2.2 Explainability analysis for invariant classifiers

While the reproduced experiments for the original paper provide some support for claim ODR, these results primarily show the effect of using counterfactuals on test accuracy performance. However, it is not directly clear from these quantitative experiments if the performance increase is actually due to the fact that the use of counterfactuals ensures that the classifier focuses on the right correlations (e.g., shape) and not spurious ones (e.g., background). To further verify the validity of claim ODR, we provide two additional analyses that combine qualitative and quantitative measures to evaluate the behaviour of the counterfactual classifiers.

**What does the latent feature space look like?**  First, we visualize learnt classifier features using t-SNE for a subset of the test set of original and counterfactual (CF) data for C-MNIST. Figure 3(a) shows that a classifier trained on CF data is indeed invariant to spurious correlations (e.g. digit color). Figure 3(b) shows that a classifier trained on CF data is also better at representing OOD samples (e.g. counterfactuals). Interestingly, the latter figure also shows that the CF-trained classifier tends to group the clusters for 4-7-9 and 3-5-8 close to each other, which was not the case for the classifier trained on original data. These digits are also close in shape in reality, which suggests that the model is rightly focusing on the shape while ignoring texture. The results for other MNIST variants are consistent with this finding.

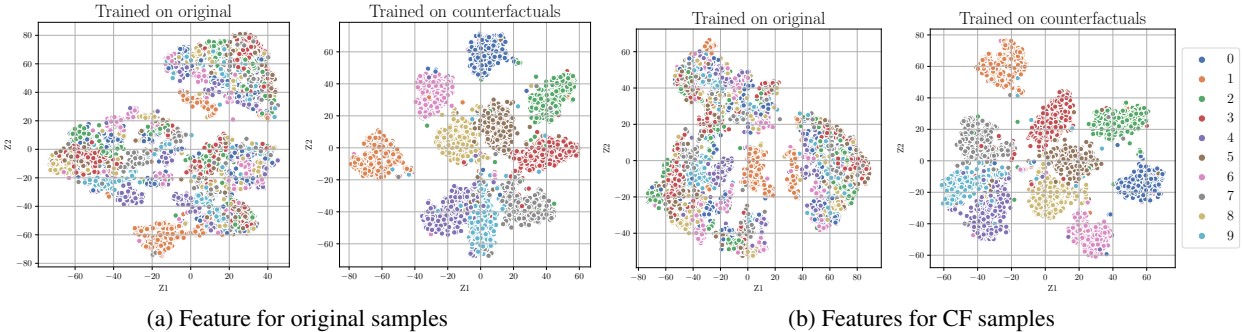

(a) Feature for original samples                  (b) Features for CF samples

Figure 3: **Feature visualization.** Feature space of a CNN classifier trained on original/CF data for colored MNIST.

**What features does the model focus on?** Second, we perform an experiment to visualize a spatial heatmap of areas that the model focuses on to make a prediction. Based on claims ODR and IBR, we would expect the different heads to operate separately from one another, while being completely invariant to the other FoVs. In order to generate the spatial heatmaps we use GradCAM. Some qualitative samples are shown in Figure 4. In addition to the qualitative analyses, using GradCAM provides the opportunity to formulate another quantitative measure to validate claims ODR and IBR. This quantitative analysis aims to measure if CF-trained models focus on shape more than those trained on original data. To this end, we compute the mean Intersection of Union (IoU) between GradCAM heatmaps and binarized digit masks on the test set. We note that a classifier trained on CF data is consistently outperforms the classifier on original data.

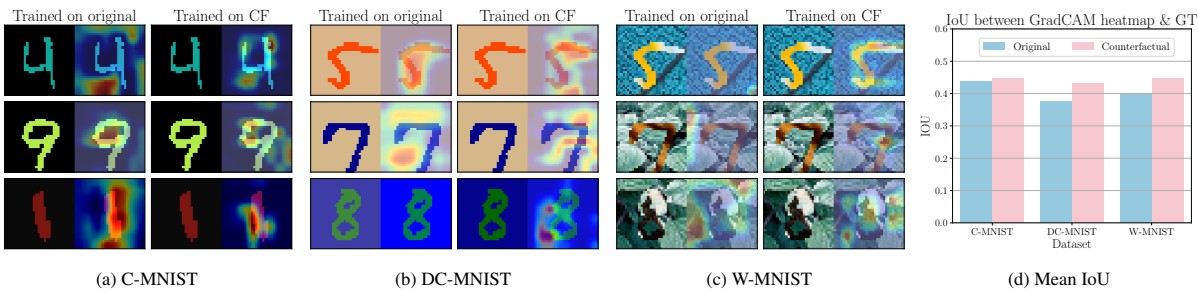

(a) C-MNIST          (b) DC-MNIST          (c) W-MNIST          (d) Mean IoU

Figure 4: **Explainability analysis:** (a) to (c): Visualization of GradCAM heatmaps for samples from each of the MNIST datasets. (d): Mean IoU between GradCAM heatmaps and ground truth binarized digit masks.

While the quantitative results using the IoU metric cannot be performed on the ImageNet data, due to the lack of ground truth binary object maps, it is possible to evaluate the qualitative performance of the independent mechanisms using GradCAM. As shown in Appendix H, the individual classifier heads tend to focus on meaningful aspects.

### 4.2.3 OOD generalization for invariant classifiers

In order to provide further evidence for the claim ODR, we test the model performance on alternative ImageNet datasets, which are designed to evaluate out-of-distribution robustness. Specifically, we evaluate the performance on ImageNet-A (natural adversarial examples) [13], ImageNet-Sketch [27] and Stylized-ImageNet [9], and compare with a ResNet-50 baseline that is pretrained on IN-1k. Surprisingly, we find that the finetuned CGN-based ensemble performs worse on all specified OOD-benchmarks, compared to the pretrained ResNet-50 baseline as shown in Table 6.

Table 6: Comparison of top-1 accuracy of invariant classifier with pretrained ResNet on OOD benchmarks.

| Model | Pretrained | Finetuned | IN-mini ⇑ | IN-A ⇑ | IN-Sketch ⇑ | IN-Stylized ⇑ |
|---|---|---|---|---|---|---|
| ResNet-50 | IN-1k | - | 75.580 | 3.400 | 24.092 | 19.218 |
| CGN Ensemble | IN-1k | IN-mini + CF | 56.793 | 1.387 | 11.775 | 17.188 |

# 5 DISCUSSION

Throughout this work, we have conducted several experiments to reproduce the main results from the research by Sauer and Geiger [22]. The results of our reproducibility study provide support for their claims, as we were largely able to reproduce the original results. Specifically, our results showed that the test accuracy for the MNIST classifiers greatly improved when using generated counterfactual datasets. Then, we were able to use the ImageNet-mini dataset to achieve similar performance trends compared to the original paper in terms of shape versus texture bias evaluation, and the background robustness evaluation. However, based on the qualitative analyses for claim HQC, it is clear that the quality of the generated counterfactual images could still be improved. Specifically, we have observed some distinct failure cases regarding the quality of generated counterfactual images, which are described in Appendix I.

Interestingly, while the loss ablation study provided similar results to what the authors reported in the original paper, we did obtain different results for the experimental run without texture loss. As the authors used this study to provide evidence for claim IBR, this difference is quite significant. Nonetheless, qualitative analysis of the images that were generated without texture loss revealed that the quality of the generated images indeed reduced when the texture loss was omitted. Although this does provide support for claim IBR, it also shows that the IS and $\mu_{mask}$ metrics used by the authors in the loss ablation study may not be sufficient to support their claims. Since the loss ablation study is therefore not conclusive, further research is required to investigate if the inductive biases introduced by the authors are indeed 'appropriate'. The results from our additional experiments provide further evidence that counterfactual images generated with the proposed CGN architecture can be used to train classifiers that are more robust against spurious signals. Using GradCAM, we were able to visualize this behaviour and formulate a quantitative performance metric.

Overall, the experiments from the original paper were largely reproducible, and their main claims seem reasonably substantiated but could benefit from additional evidence in future research. The code implementation of our reproducibility study is publicly available [1].

**Limitations** Unfortunately, we did encounter some difficulties during the reproduction process. First, since our model was trained on IN-mini, we were not able to reproduce the exact same results as the original paper. However, despite the slightly deviating results, the overall trends in the results seem to correspond well with the original results. Second, as some experimental setup information was missing from the original paper, we had to rely on the default parameter configuration files that were provided in the original code implementation, even though we can not be completely certain that these parameters were used for the original experiments.

## 5.1 Reflection: What was easy, and what was difficult?

The original paper provides an extensive appendix with implementation details and hyperparameters. Beyond that, the original code implementation was publicly accessible and well structured. As such, getting started with the experiments proved to be quite straightforward. The implementation included configuration files, download scripts for the pretrained weights and datasets, and clear instructions on how to get started with the framework.

Nonetheless, reproducing the original results turned out to be far from trivial as the setup of some of the experiments required severe modifications to the provided code. Additionally, some details required for the implementation are not specified in the paper or inconsistent with the specifications in the code (e.g., the GAN as mentioned in Section 3). Lastly, in evaluating robustness to OOD, getting the baseline model to work and obtaining numbers similar to those reported in the respective papers was challenging, partly due to baseline model inconsistencies within the literature.

## 5.2 Communication with original authors

We have reached out to the original authors to get clarifications regarding the setup of some of the experiments. For example, we asked the authors if they could share pretrained weights from the classifiers that were trained on full ImageNet, and which type of GAN architecture was used for the MNIST experiments. Unfortunately, we received a late response and only a subset of our questions was answered, and as a result we were not able to fully verify whether our design choices were consistent with those of the original paper.

---

[1]`https://github.com/anonymous-user-256/mlrc-cgn`

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

## A  Counterfactual Generative Network Architecture

In Figure 5, we provide an overview of the architecture of the CGN as provided in the paper. It illustrates how the CGN is split into four mechanism: the shape mechanism, the texture mechanism, the background mechanism, and the composer. Each mechanism takes a noise vector $\boldsymbol{u}$ and a label $y$ as input. To generate a counterfactual image, we sample $\boldsymbol{u}$ and then sample a separate $\boldsymbol{y}$ for each mechanism Sauer and Geiger [22].

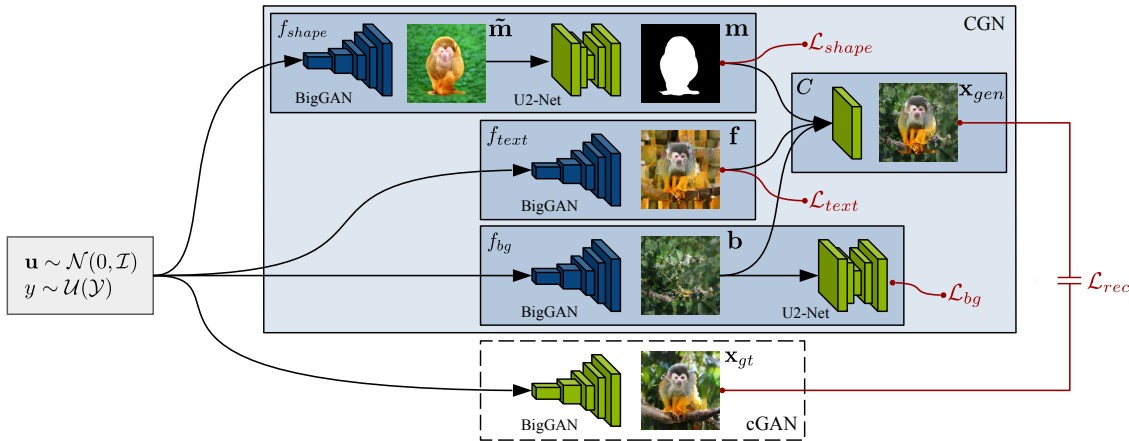

Figure 5: **CGN architecture.** Components with trainable parameters are blue, components with fixed parameters are green [22]. The dotted lines indicate that the cGAN is only used for training [22].

## B  Counterfactual images and explainability in artificial intelligence

One of the primary contributions of the work by Sauer and Geiger [22] is the proposed method to create high-quality 'counterfactual' images, which can be used to make a classifier more robust to spurious signals. As the concept of *counterfactual explanations* is closely related to the idea of explainable artificial intelligence (XAI) but is never explicitly mentioned in the paper, we first want to place the article in a broader context to achieve a deeper understanding of how the considered work relates to other developments within this field of research [3].

Based on the review by Verma et al. [26], approaches for explainability in machine learning can be roughly divided into one of two categories: (i) methods that use inherently interpretable and transparent models, and (ii) methods that generate post-hoc explanations for opaque models. The idea of counterfactual explanations belongs to the example-based approaches within the category of post-hoc explanations, that seek to offer explanations by either providing datapoints that receive the same prediction label as the observed datapoint, or by providing datapoints whose prediction label is different from the observed datapoint.

Consider the example where a classifier is trained to distinguish images from polar bears and American black bears. Given an image that has been classified by the model as a black bear, we could attempt to provide a post-hoc explanation for the model's prediction using a visual counterfactual explanation (i.e., a modified version of the input image that would be classified as a polar bear instead). These explanations can, for example, be generated using techniques such as StylEx [16]. A reasonable visual counterfactual explanation could consist of the input image, modified such that the fur of the black bear is now colored white. However, as most images of polar bears have a snow-background, and most images of American black bears likely do not, it is possible that the suggested visual counterfactual explanation still contains a black bear, but now on a snowy background.

In this case, one could argue that the background-explanation that is captured by the model is a spurious signal. That is, the classifier 'falsely' makes predictions on the background, even though the background, in reality, does not affect the actual object itself. Although this spurious signal might seem innocent within the context of this example, other spurious signals can play a role in a variety of high stake deep learning applications, such as AI in medical-imaging [5] and networks trained for military purposes [12]. While counterfactual explanations are thus capable of *revealing* such spurious signals, the proposed method using counterfactual images by Sauer and Geiger provides an approach to *mitigate* this effect.

## C    Improved CGN Training for MNIST

While training the CGN on the MNIST, we encountered an issue that was not mentioned in the original paper. During the training process, we observed that while some digits were captured almost perfectly by the model, other digit masks seemed to collapse to a state where there was a black circular shape in the center of the image with a surrounding white border (see Figure 6). When using the generated counterfactual datasets from these imperfect models to train a classifier, we then observed that the number of 'correct' (i.e., non-collapsed) images correlated strongly with the classifier performance.

Any attempt to remedy this issue using adjusted hyperparameter configurations proved to be ineffective, because the hyperparameter names in the provided default configuration-files did not directly correspond to the descriptions given in the original paper. This observation inspired a solution where we add an extra loss term to the training objective, which penalizes mask-pixels at the borders of the image. Specifically, if we define the edge region $\mathcal{E}$ as the set of pixels that are within $s$ pixels from the edge, the edge loss function can be defined as the sum of all pixel values $m_i$ within the specified edge region:

$$\mathcal{L}_{edge}(\boldsymbol{m}) = \mathbb{E}_{p(\boldsymbol{u},y)} \left[ \frac{1}{N} \sum_{i=1}^{N} m_i \cdot [i \in \mathcal{E}] \right], \tag{3}$$

where $N$ denotes the number of pixels in mask $\boldsymbol{m}$, and $[\cdot]$ denotes the Iverson bracket. As the original MNIST images in the training and test datasets often contain almost no pixels at the borders, this loss function returns values close to 0 for all ground truth MNIST images. During our experiments, we used a border size of 3 pixels, as this configuration seems to perform well to mitigate the mask-collapse issue, while still giving loss values close to 0 for the original MNIST images. By using this extra loss function, the training process became much more consistent and lead to an average classifier test accuracy of 89.8% for the Colored MNIST dataset, which is close to what was reported in the original paper.

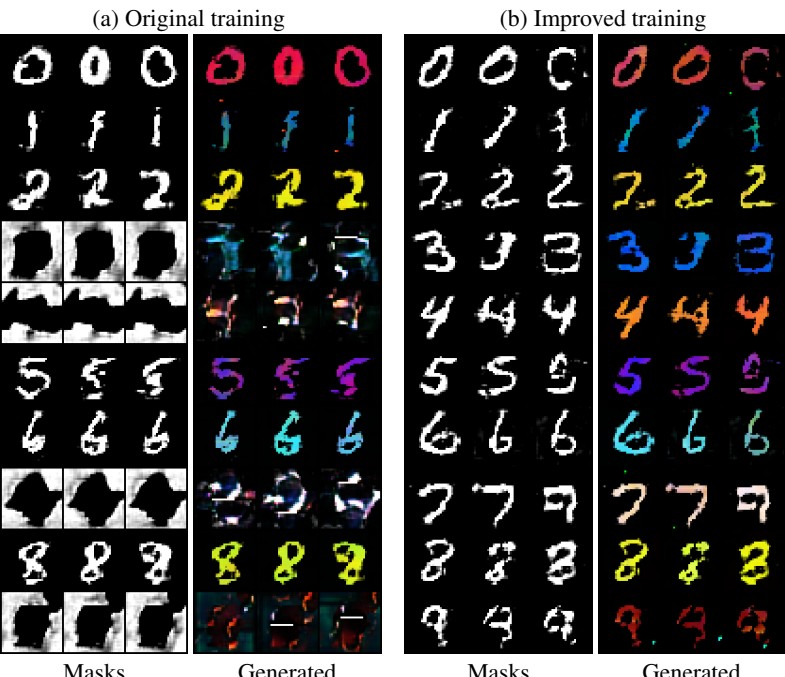

Figure 6: **Qualitative edge loss evaluation.** Adding the edge loss significantly improves CGN training on colored MNIST.

In Figure 10, we show that our modified training formulation improves the quality of generated images. In particular, we notice that incorporating $\mathcal{L}_{edge}$ in the mask loss, on average, noticeably decreases the number of non-broken images.

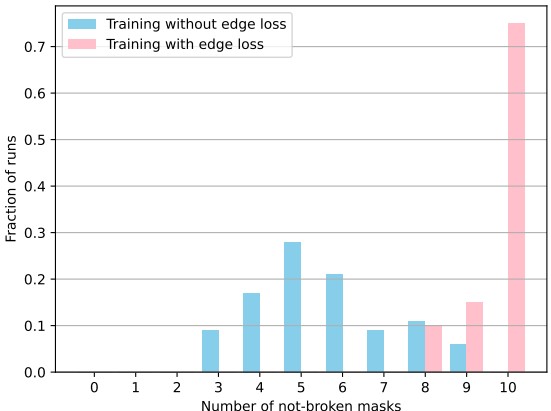

Figure 7: **Quantitative edge loss evaluation.** The fraction of experiment runs for each number of 'correct' digits.

## D Computational Cost Taxonomy

Table 7: **Cost taxonomy.** Overview of the computational cost associated with each experiment.

| Experiment type | Experiment name | Support of Claim | Section | Computational Cost (GPU Hours) |
|---|---|---|---|---|
| Reproducibility Study | Evaluating counterfactual samples | HQC | 4.1 | 0.0 |
| | Required Inductive Biases | IBR | 4.1 | 84.0 |
| | Evaluating invariant classifiers: MNIST | ODR | 4.1 | 6.0 |
| | Evaluating invariant classifiers: IN-Mini | ODR | 4.1 | 8.0 |
| | Ablation study (Appendix G) | ODR | 4.1 | 14.0 |
| Additional results | Improved CGN Training | HQC | 4.2.1 | 48.0 |
| | Explainability analysis: MNIST | ODR | 4.2.2 | < 1.0 |
| | Explainability analysis: IN-Mini | ODR | 4.2.2 | < 1.0 |
| | OOD generalization evaluation | ODR | 4.2.3 | < 1.0 |

## E Qualitative Analysis of Loss Ablation Study

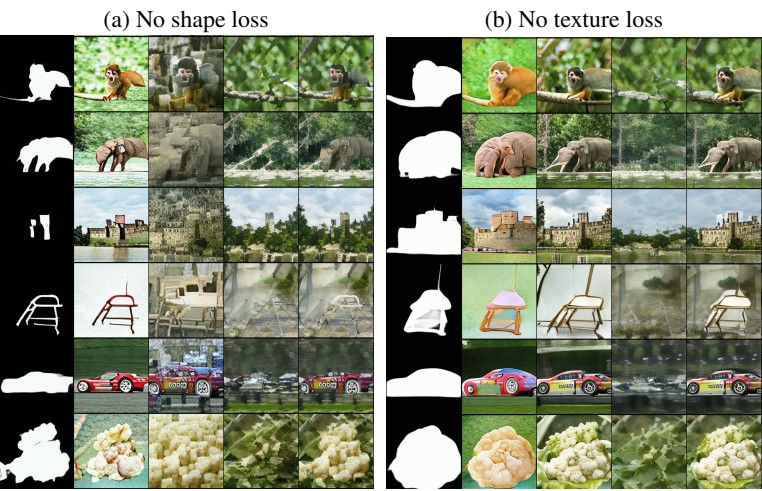

Figure 8: **Qualitative Loss Ablation.** Comparison between IM outputs when excluding the shape loss and texture loss. From left to right: $m, \tilde{m}, f, b, x_{gen}$ as described in Section 2.

## F GAN-based Baseline for MNISTs

We follow the ConvNet-based architecture for the generator inspired by PyTorch DCGAN tutorial and retain the linear discriminator as is used by Sauer and Geiger [22]. We only use binary cross entropy loss for adversarial training of both G and D. All necessary hyperparameters are same as for the CGN training. These along with pretrained weights can be found in our code repository.

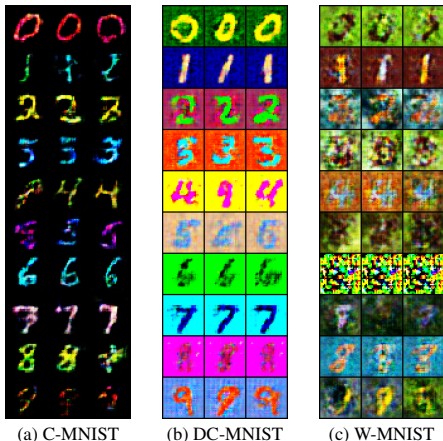

(a) C-MNIST     (b) DC-MNIST     (c) W-MNIST

Figure 9: **GAN samples.** Samples generated by a GAN baseline on MNIST variants.

## G Reproduced MNIST Ablation Study

Figure 10 shows our reproduced results for the MNIST ablation study. Our results show that using more counterfactual datapoints generally improves the test accuracy, although this was not the case for the Colored MNIST dataset, where the test accuracy decreased when using $10^6$ counterfactual datapoints instead of $10^5$. However, the difference in performance is only minor. The differences in CF ratios do not seem to have a significant effect on the test accuracies. These results seem to support the claim from the original paper that using more counterfactual images always increases the test domain results for MNIST datasets, although there only seems to be a significant performance increase when using $10^5$ datapoints instead of $10^4$. Using even more datapoints does not seem to provide a significant increase in performance.

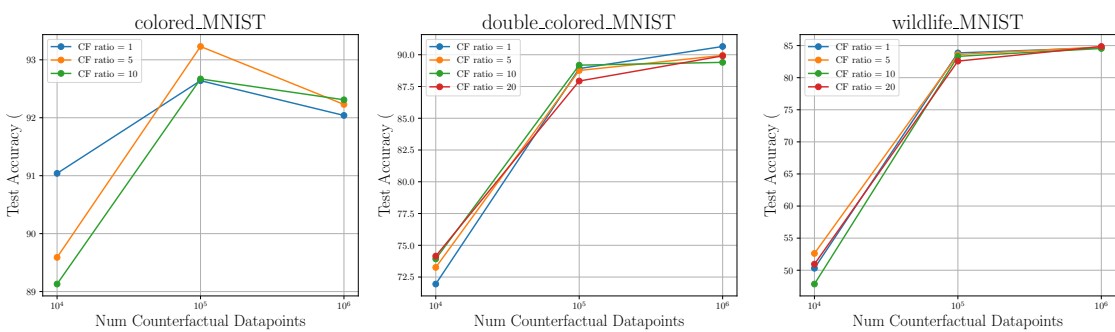

Figure 10: **MNIST ablation study.** We evaluate the impact of using more counterfactual data and generating more counterfactuals per sampled noise on the measured test accuracy.

## H GradCAM samples on ImageNet-mini

A classifier trained jointly on original and CF data is expected to have encoded invariances for certain attributes and distinctiveness for others. Recall that the proposed classifier architecture for ImageNet is an ensemble with three heads for shape, texture and background. We pose the question: What spatial aspects of an image does each head *focus* on and

what prediction does it lead to? We answer this qualitatively by analyzing GradCAM heatmaps for outputs of each of the heads as well as the averaged ensemble output. In general, the individual heads tend to focus on meaningful aspects, as shown in Figure 11, background head focuses on background. Further, for original images, we observe that a correct prediction often relies on shape (e.g., *puck* in Figure 11a) or texture (e.g., *goldfinch*). In some cases, it correctly relies on background (e.g., *castle*). For counterfactuals, surprisingly, in most cases we found that the label predicted from shape, although correct, is dominated by incorrect label from background and texture. This may be a symptom of either insufficient counterfactual training data or the use of IN-mini instead of IN-1k. We further note that texture often drives the label decision for counterfactuals.

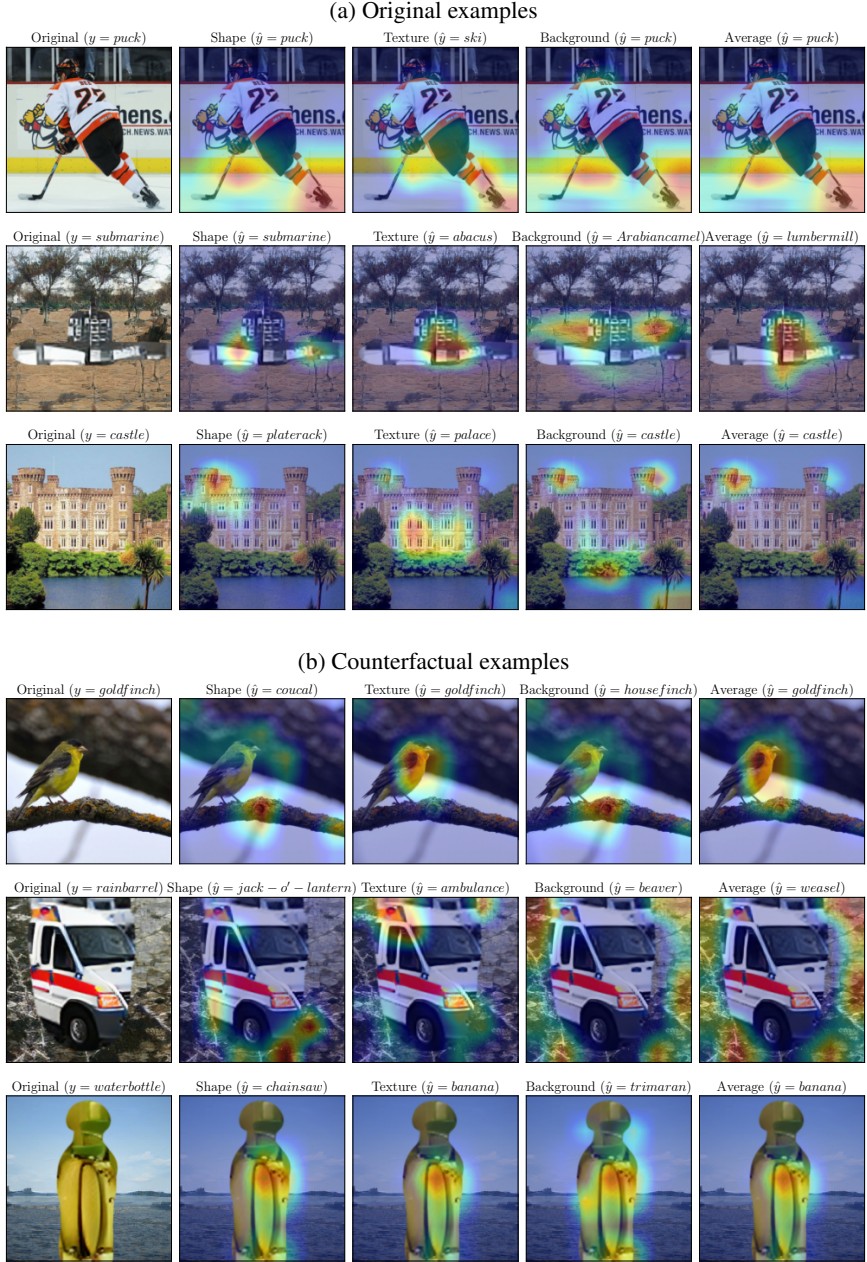

Figure 11: **Explainability Analysis ImageNet.** GradCAM heatmaps visualized with respect to individual head outputs for original and counterfactual samples. The coresponding ground truth labels and predictions are provided too.

# I Some failure modes in CGN-generated samples

Since generation of high-quality counterfactuals is one of the main claims of the paper, we perform a deeper qualitative analysis to observe if there exist typical failure modes. Based on anecdotal evidence, we note the following observations.

**Texture-background entanglement for small objects** For cases with small objects on a uniform background, such as the bird `kite in sky`, shown in Figure 12(a), or `skiing on snow`, shown in Figure 12(b), we see consistent entanglement between texture and background.

**Objects with complex texture** We observe that objects with complicated texture, such as `crossword puzzle`, shown in Figure 12(c), result in poorly recovered texture by the CGN.

**Complex scenes** As one would expect, the CGN approach does not generalize to complex scenes since it assumes a simplistic causal structure. We show an example of this in Figure 12(d).

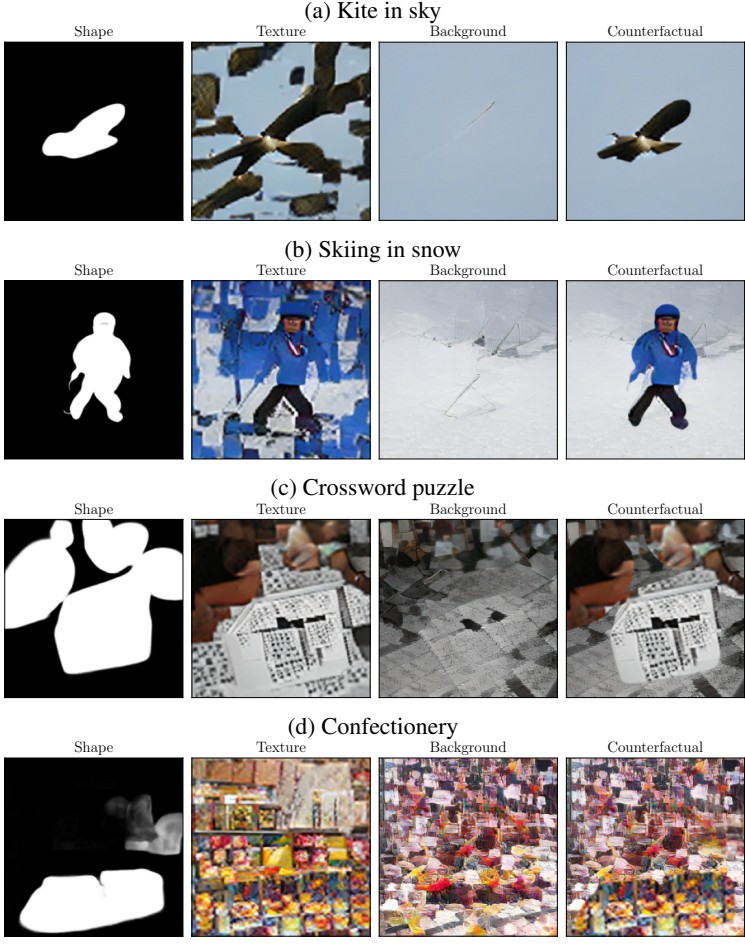

Figure 12: **Failure modes.** Cases highlighting some common failure modes in samples generated using CGN.

