# OpenReview forum: "Reproducibility Study of “Counterfactual Generative Networks”"
_ML_Reproducibility_Challenge/2021/Fall — RC2021 BestPaper_

### Official Review · Reviewer_PaFb · 2022-03-01
**Well done!**

**Rating:** 8
**Confidence:** 2

**Review:**

### Reproducibility Summary
The authors have the mandatory first page and the contents are good.

### Scope of reproducibility
The scope is clear and the evidence is provided for all claims, although some experiments were only partially reproduced due to missing specification of hyperparameters and resource constraints.

### Code
The reproducibility authors re-used the source code by the original paper authors and released their own in https://github.com/anonymous-user-256/mlrc-cgn .

### Communication with original authors
The report mentions contacting and communicating with the original paper's authors but getting a late response.

### Hyperparameter Search
The authors reused the original paper's or original codebase' values when available, and only run experiments once.

### Ablation Study
The report includes ablations and further experiments

### Discussion on results
The report discusses the results obtained.

### Recommendations for reproducibility
There are recommendations for reproducibility, mainly things that were misspecified or underspecified, and trying low-compute experiments for ease of reproducibility.

### Overall organization and clarity
Great

---

### Official Review · Reviewer_WyDQ · 2022-03-01
**The paper is really well-written and should be accepted definitely!**

**Rating:** 9
**Confidence:** 4

**Review:**

Thanks for your great paper!

Summary: The authors re-implemented the codes for most experiments and managed to validate the 3 claims of the original paper: high-quality counterfactuals, inductive bias requirements, and out-of-distribution robustness. Besides, the authors also improved CGN training on the MNIST dataset and provided a new insight of explainability analysis for invariant classifiers.

Pros:

- The paper is well written and the structure of the paper is very consistent
- The paper gives enough details about the methodology of the original paper, making it easy to read even without the original paper
- The details of hyperparameters and experimental setup are clearly stated for further reproduction
- The paper proposes an interesting view to visualize the latent feature space and model-focused features using t-SNE and GradCAM respectively
- The paper also finds that the finetuned CGN-based ensemble performs worse on all specified OOD-benchmarks
- The authors provide a detailed supplementary material

---

### Official Review · Reviewer_ELVu · 2022-03-08
**Well written review and reproducibility report**

**Rating:** 8
**Confidence:** 5

**Review:**

Authors have done a very good job in reproducing the paper - although there has been some deviations in the results achieved, the authors managed to get very similar results, and they tried to contact the authors of the original paper to get some clarifications; with mixed success. Some of the original experiments are practically impossible to reproduce, especially when scaled at the size of Imagenet.

Overall a great effort.

---

### Meta-Review · Program_Chairs · 2022-04-07

**Recommendation:** Accept (Outstanding Paper)
**Confidence:** 5

**Metareview:**

An amazing reproducibility effort, that goes above and beyond the results of the original paper, and communicates its findings in a clear and approachable way.

---

### Decision · Program_Chairs · 2022-04-09

**Decision:**

Accept (Best Paper)

**Comment:**

Following the recommendation of reviewers and meta-reviewer, the paper is accepted for ML Reproducibility Challenge 2021, and will be published in the upcoming special edition of ReScience Journal.

After several rounds of discussion, debate and incorporating recommendations from the Area Chairs and Program Chairs, the report has been granted the **Best Paper Award** of ML Reproducibility Challenge, 2021, due to its very high quality quality of all-round reproducibility effort and presentation. Congratulations!